# Crystalline metamaterials for topological properties at subwavelength scales

Simon Yves[1,*], Romain Fleury[1,2,*], Thomas Berthelot[3], Mathias Fink[1], Fabrice Lemoult[1] & Geoffroy Lerosey[1]

The exciting discovery of topological condensed matter systems has lately triggered a search for their photonic analogues, motivated by the possibility of robust backscattering-immune light transport. However, topological photonic phases have so far only been observed in photonic crystals and waveguide arrays, which are inherently physically wavelength scaled, hindering their application in compact subwavelength systems. In this letter, we tackle this problem by patterning the deep subwavelength resonant elements of metamaterials onto specific lattices, and create crystalline metamaterials that can develop complex nonlocal properties due to multiple scattering, despite their very subwavelength spatial scale that usually implies to disregard their structure. These spatially dispersive systems can support subwavelength topological phases, as we demonstrate at microwaves by direct field mapping. Our approach gives a straightforward tabletop platform for the study of photonic topological phases, and allows to envision applications benefiting the compactness of metamaterials and the amazing potential of topological insulators.

[1] Institut Langevin, CNRS UMR 7587, ESPCI Paris, PSL Research University, 1 rue Jussieu, Paris 75005, France. [2] Laboratory of Wave Engineering, Institute of Electrical Engineering, EPFL, Station 11, Lausanne 1015, Switzerland. [3] NIMBE, CEA, CNRS Université Paris-Saclay, CEA Saclay, Gif sur Yvette Cedex 91191, France. * These authors contributed equally to this work. Correspondence and requests for materials should be addressed to G.L. (email: geoffroy.lerosey@espci.fr).

Metamaterials are manmade composite media which are by definition structured at scales that are much smaller than the wavelength of operation[1–4]. As a consequence, these exotic materials are usually described by macroscopic effective properties, which can eventually be engineered at a mesoscopic scale, for instance in the context of transformational approaches[5,6]. A specific class of metamaterials, the locally resonant ones, is composed of subwavelength elements that present a resonant scattering cross-section. Without loss of generality, we hereby consider a simple resonant example in the microwave range which is the quarter wavelength metallic rod standing on a ground plane. These metamaterials are very analogous to dielectrics that consist of atoms arranged at very subwavelength scales which, when illuminated by an incoming electromagnetic radiation, scatter part of it hence participating to the total field[7]. The macroscopic collective action on the electromagnetic propagation of these atoms is usually accounted for through the index of refraction. Similarly, locally resonant metamaterials are commonly studied for their effective properties that can be very high[8], close to zero[9] or negative[10,11]. As these media properties directly stem from the resonant nature of their building blocks, the effect of spatial structuration is usually neglected or mitigated when designing metamaterials with desired properties. Metamaterials are hence commonly understood, alike dielectrics, as a random collection of resonant inclusions that present local frequency-dispersive effective properties. In our chosen example of quarter-wavelength metallic resonators, this results in high, zero, and then negative effective permittivity around their resonance frequency.

Yet, it was shown recently that since these metamaterial unit cells can present albedos very close to unity, multiple scattering can play a major role in their effective properties[12,13], a nonlocal phenomenon difficult to capture using standard homogenization schemes[14]. The resulting spatial dispersion can, for instance, turn a single negative metamaterial into a negative index one[15,16].

In this letter, we demonstrate experimentally the full potential of crystalline metamaterials, establishing the unique relevance of resonant multiple scattering to induce new and exciting properties at the deep subwavelength scale, namely, topological ones[17]. Drastically different from previous proposals to obtain photonic equivalents of condensed matter topological insulators based on Bragg interferences[18–29] or homogenized metamaterials[30–34], our approach allows for an extension of photonic topological phases down to the deep subwavelength regime, exploiting multiple-scattering and spatial dispersion.

## Results

**Subwavelength band structure engineering.** To start with, we get away from the typical description of metamaterials using effective properties of Fig. 1a,b, and adopt a solid state approach. It is known

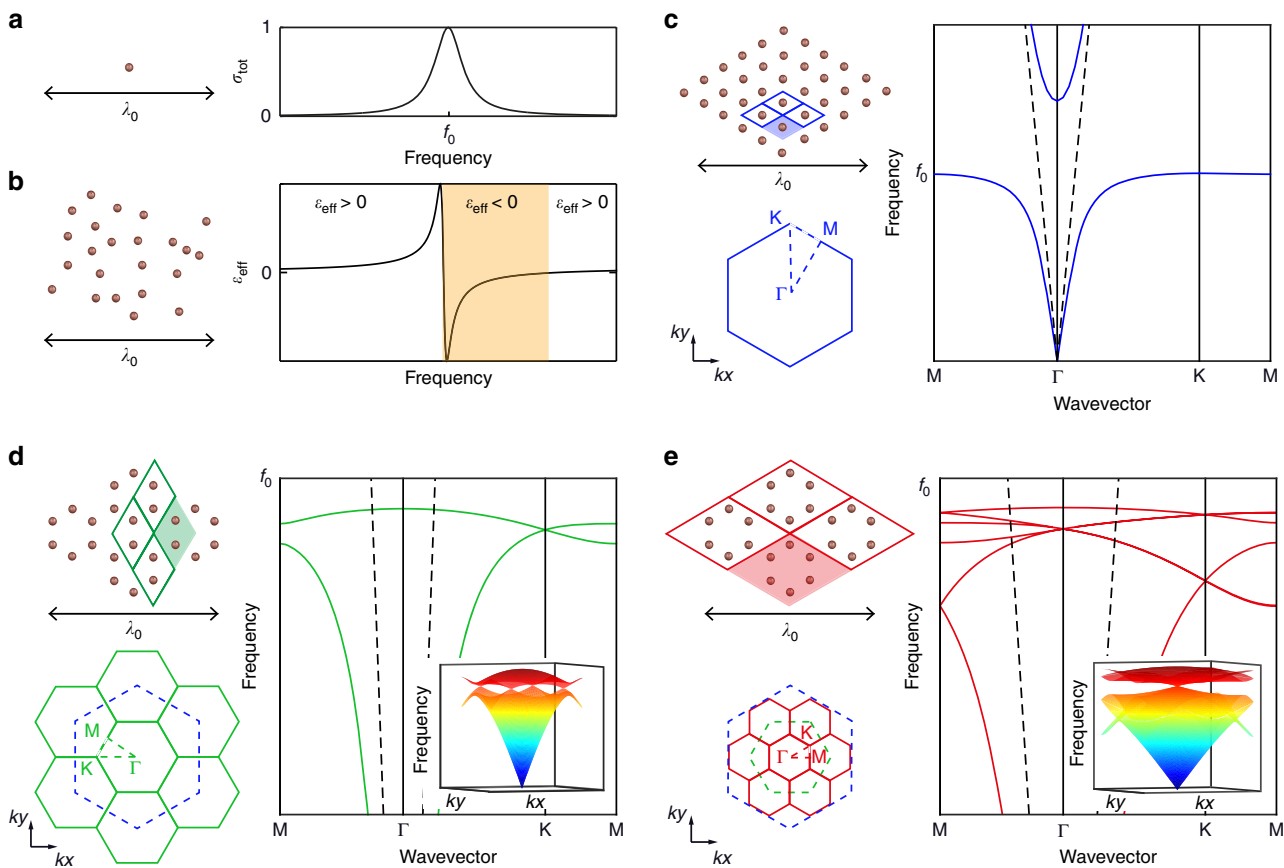

**Figure 1 | Subwavelength band structure engineering in resonant metamaterials.** (**a**) A subwavelength resonator and its scattering cross-section. (**b**) A dense two-dimensional ensemble of subwavelength resonators with arbitrary structure makes a metamaterial with negative permittivity (orange band) after the individual resonance of the resonators. Here the resonators are quarter wavelength copper rods sitting on a ground plane. (**c**) By adding a periodic subwavelength structure, in the form of a triangular array, such a metamaterial can acquire crystalline properties, involving a Bloch band structure with typical polaritonic dispersion corresponding to subwavelength bound modes localized to the array of rods. (**d**) More complex crystalline properties can be obtained, including point Dirac degeneracy at K point, by considering the polaritons supported by a honeycomb metamaterial. (**e**) Same lattice as **c** but viewed in the extended unit cell picture, allowing for mathematical folding of two time-reversed Dirac cones at the Γ point. These two time-reversed degenerate states are exploited here as the equivalent of Kramers pairs to induce topological properties at the subwavelength scale.

that a 2D-metamaterial presents the peculiar dispersion relation of a polariton, that is, an anti-crossing between the line of the waves propagating within the host medium—here air—and that of the unit cell resonance. The two branches of propagating modes—subwavelength and supra-wavelength—resulting from this level repulsion are separated by a so-called hybridization bandgap that is a consequence of the out of phase response of the unit cells right above their resonant frequency[12]. On top of this very general effect, considering spatial ordering, for instance a triangular lattice of wires (Fig. 1c), the first consequence of the crystalline nature of the metamaterial appears: the dispersion relation which links the frequency to a wavenumber can become direction dependent. However, very different from photonic crystals, such spatial dispersion is here induced at the subwavelength scale by multiple resonant scattering. In the case of simple lattices like the triangular one, this effect is weak and mostly gives rise to isotropic properties, a reason why metamaterials are usually described with effective parameters.

Now we would like to induce more crystalline effects in the dispersion relation. To do so, we move from the triangular lattice of Fig. 1c, to the honeycomb one whose unit cell contains 2 resonators, as schematized in Fig. 1d. The band structure of the new metamaterial displays more branches and can no longer be simply described in terms of local effective permittivity. Notably, as a direct consequence of the symmetry of the honeycomb lattice, a typical degeneracy of two modes appears at the **K** point: a Dirac cone. Note that the triangular lattice also supports Dirac cones at higher frequencies (outside the range represented in Fig. 1c) that are located within the light cone. Very differently, the Dirac points in the subwavelength honeycomb lattice are located well below the light cone (dashed line); Thus, the modes existing at and around this Dirac degeneracy are evanescent in the surrounding air and display spatial variations that, akin to the metamaterial typical spatial scale, are much smaller than the freespace wavelength. Importantly, this graphene-like dispersion relation results directly from the structure of the metamaterial, which enables multiple scattering to occur between the resonators despite their deep subwavelength spacing[15]. Interestingly, this multiple-path propagative coupling, which creates the polaritonic dispersion, is totally different from the typical tight-binding electronic coupling occurring in a graphene sheet, which would lead to sinusoidal branches.

Going from the triangular lattice to the honeycomb one can be pictured as a folding of the band structure, the area of the first Brillouin zone being divided by 3 (Fig. 1d). To start off ideal conditions for inducing topological properties, we take inspiration from an idea initially proposed by Wu and Hu in the context of non-resonant photonic crystals[26] and fold again the obtained band structure, creating additional point degeneracies. Thus, we consider the supercell containing a full hexagon of resonators (Fig. 1e) instead of the primitive one containing only the 2 resonators (Fig. 1d). The band structure is now a mathematical folding of the previous one i.e., it has no real physical meaning but it is merely used as a preliminary trick to introduce a topological behavior in the system. In this interpretation of the honeycomb structured metamaterial, multiple scattering occurs between hexagons arranged on a triangular lattice, and the two Dirac cones previously located at $+\mathbf{K}$ and $-\mathbf{K}$ are now folded into the $\Gamma$ point, thereby resulting in a fourfold degeneracy at the center of the first Brillouin zone. It is worth pointing out that although the fourfold degenerated modes at the double Dirac degeneracy are located at the $\Gamma$ point, they are still evanescent and hence of deep subwavelength spatial scale. Indeed, we have solely mathematically folded the dispersion relation of the metamaterial structured on a honeycomb lattice, meaning that very high wavenumber modes

have fallen into the first Brillouin zone, yet still keeping their evanescent nature.

The presence of this double Dirac cone is of primary importance to induce nontrivial topology. Indeed, a single Dirac degeneracy is not sufficient to obtain topological properties in a time-reversal invariant system and the presence of degenerate time-reversed Kramers pairs is also required, or in other words, the band structure should feature two overlapped, time-reversed Dirac cones, as those obtained in Fig. 1d. For fermions, for which the time-reversal operator squares to $-1$, this condition is automatically fulfilled due to Kramers theorem[17]. Conversely, for bosons or classical waves, the time-reversal operator squares to $+1$. In systems below three dimensions, this prevents topological properties to be protected by time-reversal symmetry alone. In order to circumvent this obstacle, it is possible to design an effective fermionic time-reversal operator, augmenting time-reversal symmetry with another symmetry operation, such as electromagnetic duality[22,25], inversion[28] or rotational symmetry[26]. In this way, as long as this additional symmetry is preserved, we can use the same formalism as for fermionic time-reversal invariant system and construct Kramers pairs. Here, following the seminal proposal of Wu and Hu[26], we have exploited the six-fold rotational (C6v) symmetry of a triangular lattice, by considering a honeycomb lattice but with a supercell. We now show that very simple C6 compatible modifications of the honeycomb structured metamaterial can provide a physical folding of the band structure and result in the opening of a topological band gap, which allows the emergence of subwavelength photonic states with non-zero topological charge.

**Topological phase transition**. Two types of structural deformations are considered here, and represented in Fig. 2a,b (the wires are viewed from the top). In panel a, the size of the hexagons within the previous supercell is shrunk compared to the undeformed honeycomb lattice (transparent resonators). We obtain a new metamaterial built from hexagonal clusters of resonators, circled with green dashed lines. These building blocks are referred as metamolecules: they provide the resonant modes that interact through multiple propagative coupling to create the photonic bands of the medium. In Fig. 2b, we consider the opposite situation of increased sized hexagons compared to the undeformed situation. In this case, the resonators are brought together into different clusters, and although the lattice is built from a periodic arrangement of large hexagons, the natural backbone metamolecule to consider has a different shape and symmetry: it involves a trimer of resonator pairs, belonging to three different hexagons, circled again in green dashed lines.

To understand how these deformations modify the mathematically folded band structure of Fig. 1e, it is first instructive to look at the resonant modes of the individual metamolecules, shown in Fig. 2c,d. Indeed, the band structure of the designed metamaterial crystals can be interpreted as the hybridization between these eigenmodes and the plane waves propagating in air, in the same way as the simple triangular lattice of wires of Fig. 1c. Because of the propagative coupling between the resonators, modes with higher spatial variations along the curvilinear path of the considered metamolecules are more energetic (Fig. 1c), that is, occur at higher frequencies: therefore, the eigenmodes of the hexagonal metamolecule feature first a monopolar mode, followed by two dipolar ones, two quadrupolar modes, and a hexapolar mode (Fig. 2c).

Moving from a single metamolecule to a metamaterial crystal, the discrete eigenmodes become a complete band structure. Figure 2e shows the case of the lattice of shrunk hexagons (thick lines), and compares it to the case of the undeformed honeycomb lattice (thin lines) previously shown in Fig. 1e. The

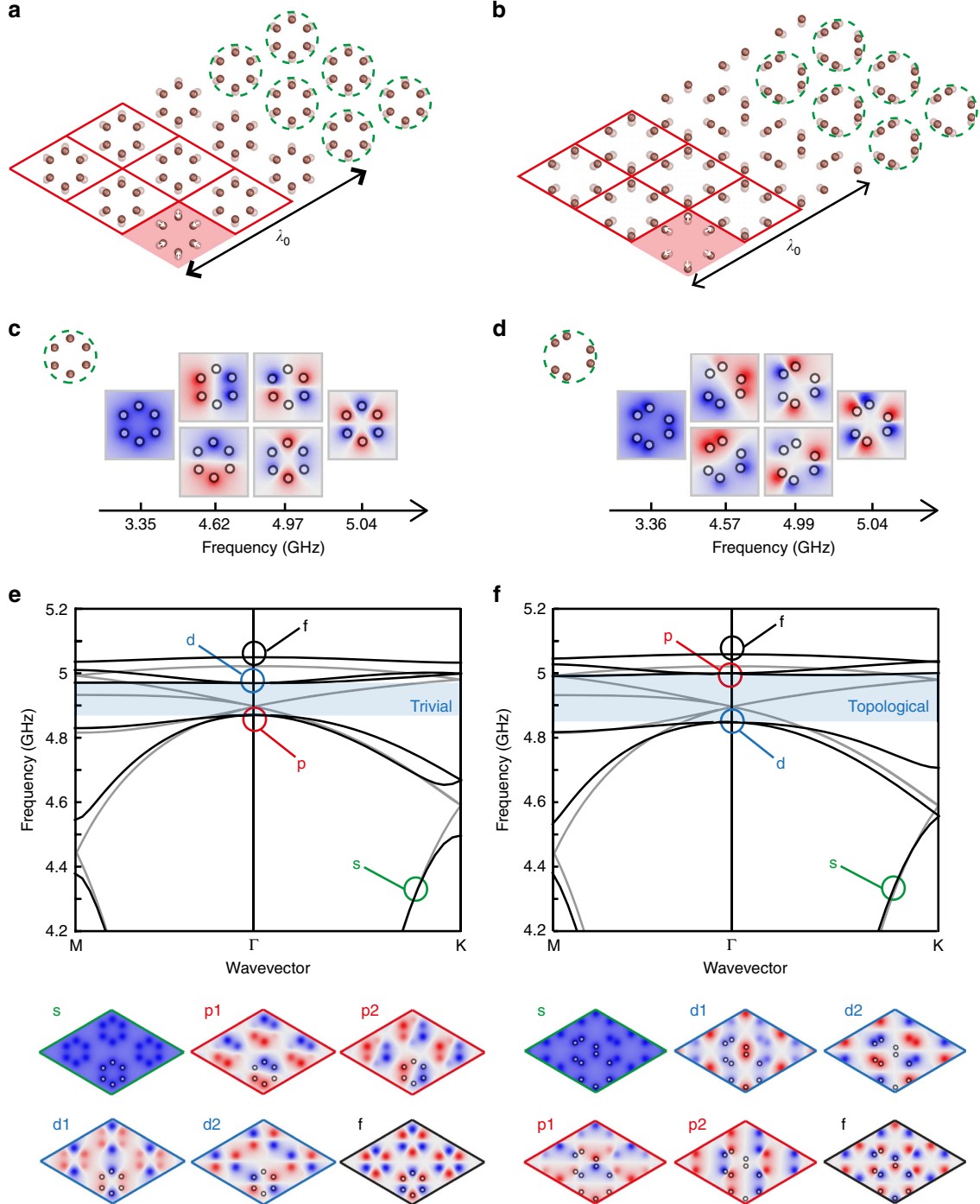

**Figure 2 | Opening of a topological band gap by subwavelength structural modifications.** (**a**) Starting from the honeycomb lattice in the extended unit cell picture of Fig. 1e, we shrink the size of the hexagonal arrangement of resonators composing each unit cell, obtaining a triangular lattice of hexagonal metamolecules (circled in green). Conversely, in **b** we expand the size of the hexagons, obtaining an array composed of a different metamolecule involving a trimer of resonator pairs (circled in green). (**c,d**) The resonant modes of the two metamolecules, responsible for the six bands observed in the band structures of the two crystals (**e,f**). (**e**) The band structure of the lattice of shrunk hexagons, and the associated electric field distribution for the Bloch modes of the six bands at the Γ point. (**f**) Similar to **e** for the lattice of extended hexagons, except that the p and d bands are now inverted with respect to the bandgap. As demonstrated in the text, the bandgap (blue shaded area) in **e** is topologically trivial, whereas the one in **f** is associated with a non-trivial topology.

previous mathematical fourfold degeneracy at Γ disappears and we notice the physical opening of a complete band gap at this frequency. Because of the C6 symmetry, the top and bottom bands remain doubly degenerate at Γ (ref. 26). Consistent with the metamolecule eigenmodes ordering, the band structure hence starts at low frequency with a band built on the monopoles

(s type). Next, the two degenerate dipolar modes of the hexagonal cluster form a pair of bands ($p_1$ and $p_2$). The next bands are due to the quadrupolar resonances of the hexagons ($d_1$ and $d_2$). Finally, the last band is due to the hexapolar mode ($f$). The lattice field distributions at Γ showing all of these geometries are represented below the dispersion relation.

Interestingly, the case of the expanded hexagons comes with a twist. A similar complete band gap appears at $\Gamma$, however the band structure is now inverted around the gap: the two bands below the gap are now of $d$-type, whereas the two bands above are of $p$-type (Fig. 2f). Such a band inversion follows from the eigenmodes ordering of the metamolecule consisting of a trimer of resonator pairs shown in Fig. 2d, which is again dictated by the propagative coupling between the resonators. For each band, the resonant mode profile on the metamolecule (Fig. 2d) construct the $p$ and $d$ symmetries on the expanded hexagons (see field distributions in the bottom panels of Fig. 2f). These mode profiles highlight how dipolar modes on the trimer metamolecule create quadrupolar modes on the hexagon, and vice versa. This simple picture explains the inversion or the order of appearance of the $p$ and $d$ bands in this second case.

To further determine the topology of these band gaps, we have computed numerically the spin Chern numbers associated with the bands $p_1 \pm ip_2$ and $d_1 \pm id_2$ (see Methods). The latter correspond to positive and negative angular momentum of the out –of –plane electric field, and can be considered as the pseudo-spin analogues of our system. We find that the band gap is topologically trivial in the case of the shrunk hexagons, whereas the case of the expanded hexagons is associated with spin-Chern numbers of $+1$ for the $p_1 + ip_2$ and $d_1 - id_2$ bands, and -1 for the $p_1 - ip_2$ and $d_1 + id_2$ bands. This confirms the topological nature of the associated band gap. Consequently, a topological transition happens exactly at the undeformed honeycomb lattice, demonstrating the drastic effect of C6 compatible deformations on the topology, even at the deep subwavelength scale of metamaterials. We will now move on to an experimental demonstration of these observations.

**Experimental demonstration of subwavelength band inversion**. Figure 3a,b show pictures of the fabricated samples of topologically trivial and non-trivial metamaterials (see Methods for the sample fabrication). The total size of the samples is comparable with the wavelength in freespace $\lambda_0$, pointing out the deep subwavelength nature of the metamaterial crystals. To observe the topological band inversion process, we excite both samples locally in their close vicinity with a homemade antenna (see Methods) and measure the transmitted electric field right above the resonators using a network analyzer. The latter allows us to make a broadband measurement in one step with a high frequency resolution. We carry out a scanning of the full area of the metamaterial. The frequency spectrum of the field amplitude, averaged spatially over a large area corresponding to 6 unit-cells in the center of the bulk, is shown in Fig. 3c (trivial sample) and Fig. 3d (topological sample). As predicted by the band structures of Fig. 2e,f, we observe zero field amplitude in the band gaps, between 4.9 and 5 GHz (shaded area). Outside these frequency regions, peaks in the spectrum traduce efficient coupling between the excitation and a bulk stationary mode. Here we notice the presence of several discrete peaks, in the frequency range corresponding to the continuous band obtained numerically. This discretization of the spectrum comes from the finite size of the sample. In the figure, we therefore include the experimental field maps associated to various frequency peaks, which correspond to different bulk bands in Fig. 2e,f. The insets located at the top right corners of the field maps reproduce the field profile measured over a given hexagon in the sample, allowing the classification of the modes into $s$, $p$, $d$ and $f$ types. By allowing for direct mapping of the bulk field distribution, our experiment unambiguously demonstrates the band inversion phenomenon. This, associated with the computed topological invariant, provides a direct confirmation of the distinct topological properties of the two samples. We also note here that the results of the finite-element

simulation for the infinite crystals (Fig. 2) are in excellent quantitative agreement with the experimental results, since the modes we measure and compute possess the same symmetries at the same frequencies. The minor deviations between the measured field maps and the computed eigenmodes of the crystal are due to the losses inherent to the metallic wires in the sample, that lead to peak broadening and weak, but noticeable, mode overlapping.

**Experimental study of subwavelength topological edge states**. A striking consequence of the fundamental topological difference between the designed media is the occurrence of topological edge modes when these samples are put into contact. As represented in Fig. 4a, we connect the two armchair edges of the experimental samples to create an interface, and excite it with a small antenna placed close to the bottom of the sample, in the axis of the domain wall. Looking at the spectrum of the electric field measured directly above the wires along the interface (Fig. 4b), we clearly observe two peaks within the frequency region corresponding to intersection of the band gaps of both bulk media (shaded zone). These peaks are symptomatic of the presence of two edge modes along the interface, one at lower frequency (LF) and the other one at higher frequency (HF). Here, because of the breaking of C6 symmetry at the interface, the edge states are gapped. Again, our experimental setup allows us to map the field distributions excited at these frequencies, which are shown in Fig. 4c,d, and confirm the presence of edge states. We compare these maps with the prediction of a semi-analytical model solving the multiple scattering of an ensemble of coupled dipoles (Fig. 4e,f, see Methods for details of the model). We obtain good agreement between our experimental and semi-analytical maps. Remarkably, the field emitted by the source efficiently couples to the interfacial modes due to the interaction between their shared symmetries. It also seems that the guided mode radiates energy towards far-field when reaching the top-end boundary, thanks to good matching between the resonant dipole located at the end and the free-space solutions. In stark contrast, leaky wave out of plane radiation is much less efficient along the interface due to the subwavelength nature of the edge mode, even if the mode enters the light cone. More precisely, the modes of the waveguide have a strong spatial Fourier component in the second Brillouin zone, since built on folded bands, and therefore the leaking rate for the edge mode is very small compared with the leaking rate at the end of the sample, which is the one associated with a matched dipole. The possibility to couple energy in and out of the interface at its ends could provide a way to exploit the unique properties of these subwavelength modes directly from the far field. Again, we emphasize that the total size of the samples is about one wavelength, so this demonstration corresponds to a subwavelength topological routing of the electromagnetic energy.

Carefully looking at the local field of the two modes reveals very different local symmetries which we now investigate by simulating an infinite interface. We evaluate the dispersion relation of potentially existing edge modes by applying Bloch boundary conditions. As shown in Fig. 4g, in the frequency range corresponding to the intersection of the two bandgaps of the bulk metamaterials (non-shaded area), two modes appear that do not exist when connecting two trivial samples or two topological ones. These modes are one-way propagating at frequencies in the close vicinity of the Gamma point. Small deviations between the frequencies obtained from FEM simulation and experiment are attributed to the imperfect alignment of the two domains in the experiment, due to geometrical constraints. The field distribution associated to each mode explains their existence

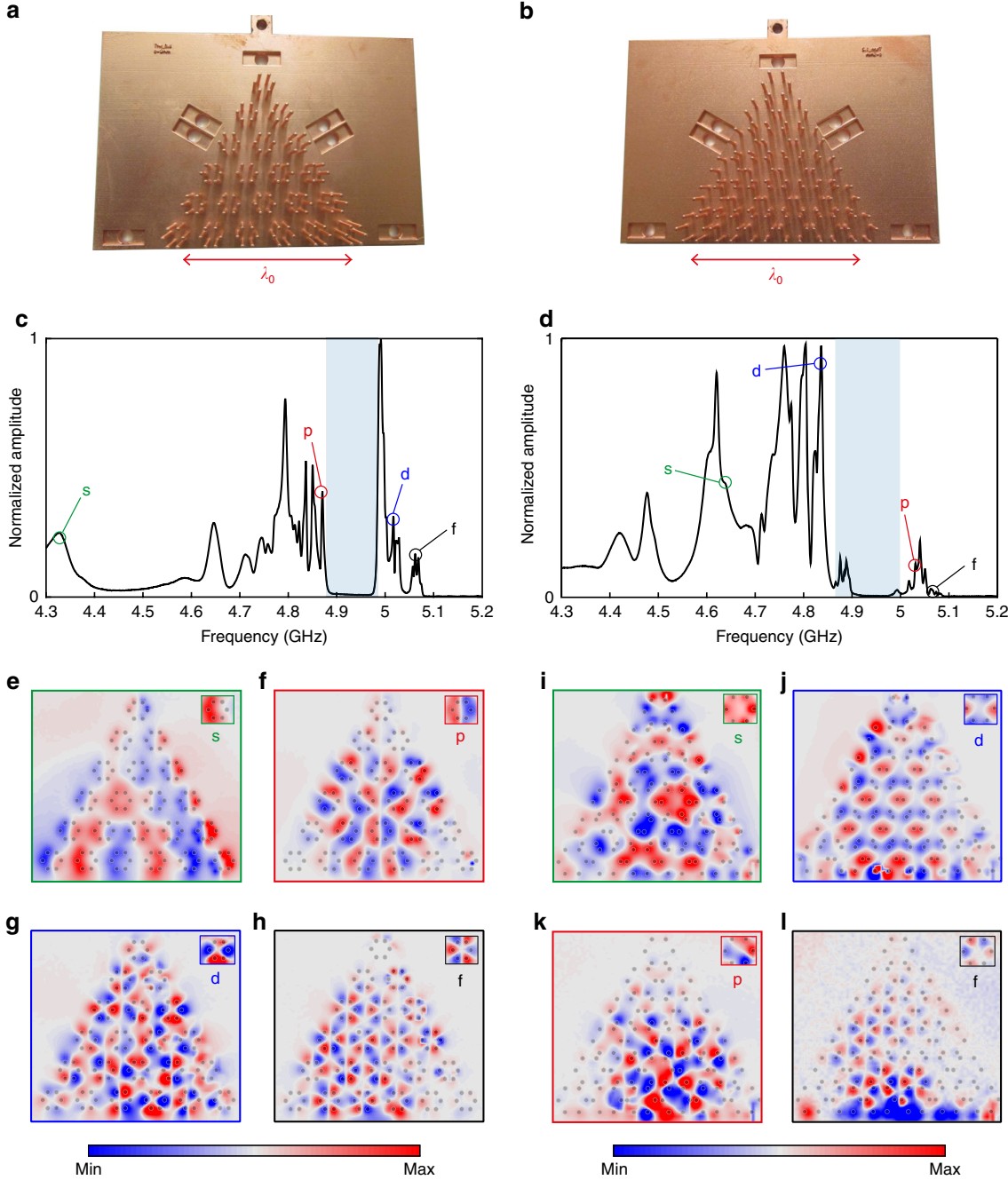

**Figure 3 | Experimental validation of subwavelength, topologically nontrivial phases.** We have fabricated by 3D printing followed by chemical metallization two subwavelength sized samples composed of conducting rods on a ground plane. (**a**) Picture of the topologically trivial sample made of shrunk hexagons. (**b**) Picture of the topologically nontrivial sample made of expanded hexagons. (**c**) Electric field amplitude spectrum in the bulk of the topologically trivial sample (spatially averaged). A complete band gap is observed (shaded frequency range). Outside of this gap, peaks correspond to efficient excitation of s, p, d and f type bulk Bloch modes, as confirmed by the electric field maps at these frequencies (**e,f,g,h**). (**d**) Same as (**c**) for the topologically nontrivial sample, demonstrating the topological band inversion phenomenon induced by subwavelength structural deformations, with the order of appearance of the p and d bands reversed with respect to the band gap. (**i,j,k,l**) Same as (**e,f,g,h**) for the topologically non-trivial sample.

with respect to the bulk properties of each medium creating the interface. Indeed, the lower frequency interface states connect the lower frequency bulk bands of each metamaterial, and hence they share both of their symmetries: they are a mixture of the *d*-symmetry of the topological medium on the left and the *p*-band of the trivial medium on the right (see inset). Conversely, the upper interface band that binds the higher frequency bulks of both metamaterials, is made out of modes that are mainly a mixture of *p* modes of the topological medium and *d* modes of the trivial one (see inset). When the two samples have the same topology, those symmetry connections appear for frequencies within a propagating band of one of the two bulks, preventing the occurrence of an edge mode in the gap. On the contrary, thanks to the band inversion, the interface connecting a topological sample to a trivial one always creates modes within the bandgap, explaining the subwavelength confinement along the interface.

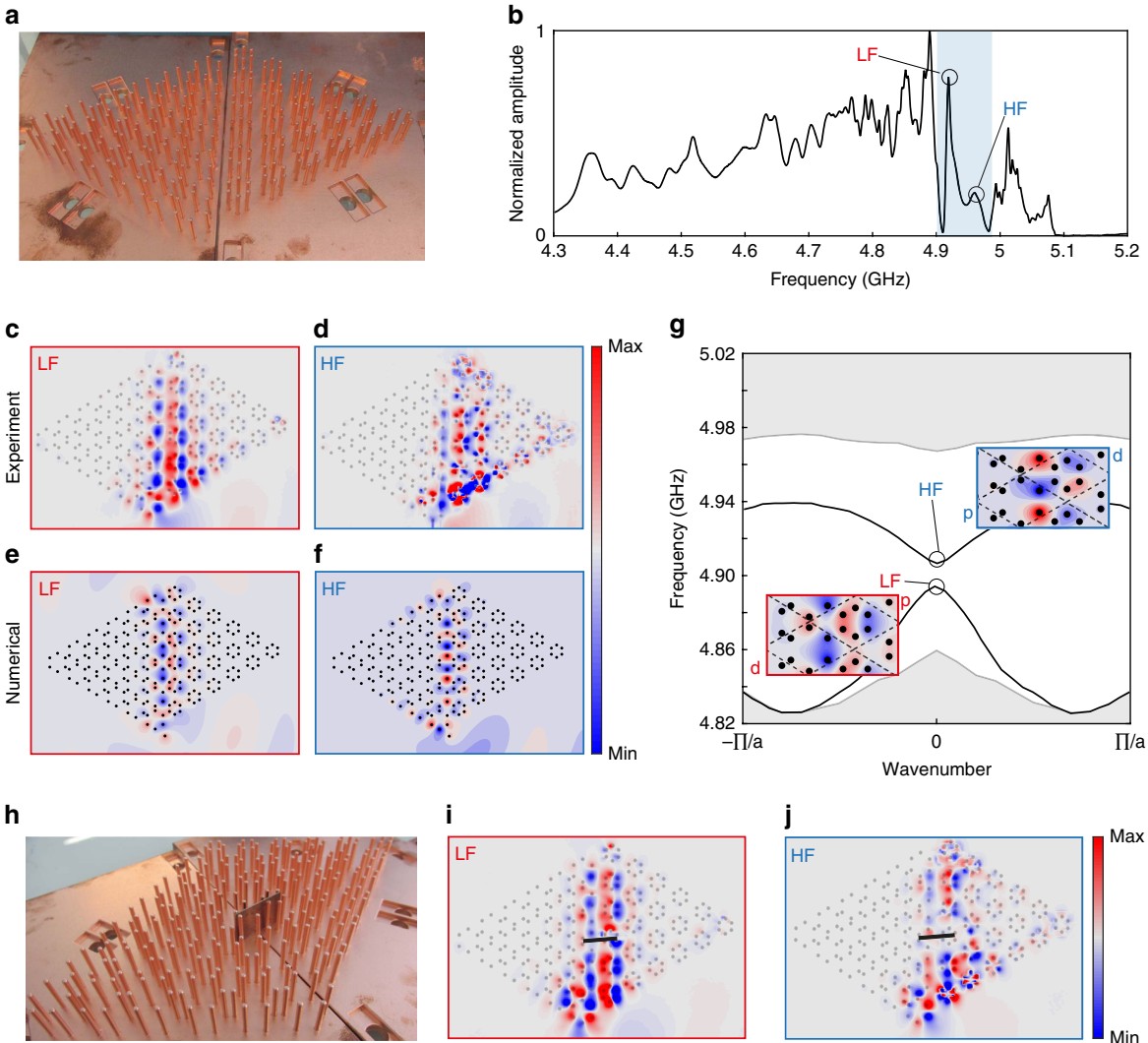

**Figure 4 | Experimental observation of topological edge modes guided at the subwavelength scale.** (**a**) We form a boundary between the topologically trivial sample and the topologically nontrivial one. (**b**) Spectrum of the field amplitude measured in the sample, averaged along the boundary. Two peaks appear in the band gap corresponding to a low frequency (LF) and a high frequency edge mode (HF). (**c,d**) Experimental electric field maps for the LF and HF edge modes. (**e,f**) Corresponding field maps computed from a coupled-dipole semi-analytical model. (**g**) Corresponding dispersion diagram for the edge modes, shown in inserts. (**h**) We insert a defect in the form of a large conducting wall placed right on the boundary. (**i,j**) Measured electric field maps in the presence of a defect, showing good robustness of the edge modes despite the breaking of sixfold rotational symmetry along the interface.

Eventually, because of the breaking of the C6 symmetry induced by the interface, these topological edge modes (topological in the sense that their existence is due to the different bulk topologies) are not topologically protected, and are potentially sensitive to backscattering. However, as the interface we designed is associated with a relatively weak breaking of the C6 symmetry, the topological edge modes propagation, although not unidirectional, remains very robust. In order to demonstrate this, we induced a defect along the interface by positioning a piece of copper plate transversely to the guiding direction (Fig. 4h). Despite the presence of such a conducting wall, the measured field maps (Fig. 4i,j) are identical to the ones obtained without the defect, demonstrating qualitatively the inherent robustness of the edge states even in the presence of stringent, C6 incompatible defects. Notably, the field amplitude is the same before or after the defect, confirming the absence of strong Fabry-Pérot interferences within the sample. Note that we could play other tricks to minimize the coupling between the two modes at the C6 symmetry breaking interface. For instance, we could consider,

instead of an abrupt interface, an adiabatic deformation between one sample to the other. Locally, the deformation of the crystal would be so slow that the sample would not feel at all the breaking of C6 symmetry. The edge modes would then be even more robust. In addition, they would retain their main interesting features: localization to the surface, ultraslow propagation and subwavelength confinement. In the methods, we further investigate the robustness of the edge modes in the presence of subwavelength turns along the interface, confirming the excellent resilience of the edge modes.

## Discussion

To conclude, in this Letter we have demonstrated the full power of crystalline metamaterials, which can be judiciously structured to induce very complex properties at the subwavelength scale. Our proposal not only allows us to demonstrate a new form of topological wave propagation at the subwavelength scale, but also highlights a very fruitful and general approach that can be

exploited to observe the metamaterial analog of many exciting condensed matter systems. Furthermore, while subwavelength topological properties are demonstrated here in the microwave range with quarter-wavelength resonators, we have tested it for acoustic metamaterials made out of Helmholtz resonators and for all dielectric metamaterials composed of resonant Mie particles, obtaining similar results; together with the coupled dipole simulations presented in the Methods section, this asserts the broad generality of the proposal. The method may be directly extended to acoustic[35–39] or mechanical[40] topological insulators, to obtain topological vibrations at the subwavelength scale. Finally, we would like to emphasize that none of these results would have been easily foreseeable considering metamaterials for their local effective properties only: it demonstrates the unique relevance of a crystalline approach of metamaterial science that fully considers the effect of multiple scattering at the level of the subwavelength structure. This suggests a new era where spatial dispersion can be fully controlled and engineered together with frequency dispersion to unleash the full potential of metamaterials, and make disruptive advances in the ability to control waves down to the subwavelength scale.

## Methods

**Experimental samples.** The samples are three-dimensional (3D) printed in an ABS-like polymer resin using an Objet30 Pro 3D printer. The surface of the polymer is first annealed then submitted to an acidic treatment to produce chemical functions for the copper metallic ions chelation. The latter are chemically reduced to create copper nanoparticles or clusters onto the surface that acts as a seed layer catalyzing the copper electroless plating. To reach the copper bulk electrical properties and hence minimize the dissipation of waves in the polymer matrix, the thickness of the copper electroless plated layer is further increased over 20 μm by performing copper electroplating of the resulting conductive samples.

**Experimental set-up.** We conduct spectral measurements using a network analyzer (Agilent Technologies N5230C) for frequencies ranging from 4.3 GHz to 5.3 GHz. It allows to make a broadband measurement (1 GHz) in one step, whose peak bandwidth is 625 kHz. It is by far sufficient to resolve any physical peak of the sample. Indeed, losses inherent to the metallic wires impose a lifetime below the microsecond, which is smaller than the 1.6 μs corresponding to the peak resolution we have. In particular for the domain wall experiment, this bandwidth of excitation equals to 0.3% of the small bandgap between the two edge modes LF and HF. The experimental setup is displayed in Supplementary Fig. 1. The system is fed by a magnetic source (a small current loop of 2 mm diameter) placed close to the bottom of the wires where the magnetic field is maximum, for optimal coupling. For the experiments on the two triangular samples (Fig. 3 in the main text) the source is set at the middle of the lower side of the triangle. Concerning the domain wall experiment (Fig. 4 in the main text), it is placed along the axis of the interface at the bottom of the samples. We probe the electric field with a homemade antenna consisting of a very short wire, so inefficiently radiative to preferentially probe the evanescent field. This process prevents from directly measuring the signal emitted by the source and therefore guarantees a better acquisition of the metamaterial's response. This probe, mounted on a 2D translational stage (Newport M-IMS400PP), scans the electric field above the whole sample around 1 mm away from the top of the wires, with a step of 1 mm. The measured spectra of each of the triangular sample we present in Fig. 3 are averaged on an area which covers six unit cells in the center of the sample to avoid boundary effects. Concerning the domain wall experiment (Fig. 4 in the main text), the spectrum is averaged on an area containing one unit cell from each side of the interface and centered in the middle of the domain wall. This is again done in order to average out the effect of the boundaries of the samples with air.

**Bloch band structure calculations.** The band structures presented in the main text were obtained by three-dimensional full wave finite-element simulations using the eigensolver of Comsol Multiphysics, RF module. The wires were modeled as perfect electrical conductors, and the crystal unit cells were surrounded by periodic boundary conditions.

**Semi-analytical model.** We have developed a semi-analytical model to demonstrate the generality of our results and investigate the physics associated with finite-sized samples. The model is based on a two-dimensional coupled dipole method[41,42], in which each two-dimensional dipole $p_i$ is modeled by its electrical polarizability $\alpha_i$, following a Lorentzian model fully taking into account the optical theorem (energy conservation)[43]. Dipoles can be excited using an external source electric field $E_i^S$, and are coupled to each other using the free-space Green's functions $G_{ij}$ linking the field created at the location $i$ by the two-dimensional

dipole located at the position $j$:

$$\alpha_i^{-1} p_i - \sum_{j \neq i} G_{ij} p_j = E_i^S. \tag{1}$$

In the calculations, only the source dipole is excited and is associated with a non-zero source field $E_i^S$. The linear system above is solved at each frequency by direct matrix inversion implemented in a C code based on the LU decomposition with partial pivoting and row inter-changes[44] implemented in the lapack package from IBM company. The electric field is then calculated at each frequency by summing all the individual contributions of all the dipoles. These simulations can incorporate losses by adding inelastic losses to the radiative losses already present in the polarizability model.

**Calculation of the spin Chern numbers.** The spin Chern numbers of the bands $p_1 \pm ip_2$ and $d_1 \pm id_2$ are calculated using the method of Fukui, Suzuki and Hatsugai[45] for fast calculation of topological invariants over a roughly discretized (15 points by 15 points) portion of the Brillouin zone around its center. The eigenmode profiles are taken directly from our 3D finite-element simulations performed with Comsol Multiphysics.

**Bulk semi-analytical simulations.** We reproduce the experiments on both topological and trivial samples with our semi-analytical model. Dipoles are displayed on the exact same lattice nodes as the wires in the experiments. The source is placed 8.5 mm below the lower side of the triangle, at the middle. The results are presented in Supplementary Fig. 2. The transmission spectra are calculated at one point in the center of the sample. As their experimental counterparts, the simulated trivial and topological samples present transmission peaks below $f/f_0 = 1$. Moreover, within this frequency range, there is a zero-transmission window centered on $f/f_0 = 0.965$, which is larger for the topological sample. The latter, shaded on the figure, corresponds accurately to the bandgap observed experimentally.

We now take a closer look at the calculated modes corresponding to the different transmission peaks in order to look at their symmetries. Those modes are represented, with a zoom-in showing their symmetry, below the spectra Supplementary Fig. 2. In the case of the trivial sample simulation, the modes appearing at the lowest frequency have the s-symmetry. Next comes the p-modes before the bandgap opening. After the bandgap closing the modes have a d-symmetry followed by an f-symmetry for the highest frequency peaks. For the topological sample simulation, the d-modes corresponds to the peaks before the bandgap opens, and the p-modes occurs when it closes. This, added to the great similarities between the calculated modes and the experimentally measured ones, is a strong proof of the band inversion that happens between the p and the d bands in any metamaterial composed of resonant scatterers and for which near field inductive or capacitive couplings can be safely neglected.

**Experimental observation of edge states between metamaterial and free-space.** In the main text we present the experimental results concerning the bulk behavior of the two types of metamaterials. Here we focus on what occurs at their edges. The results are displayed on Supplementary Fig. 3. In both cases, the spectra are averaged along the triangular contour of the sample and the shaded area corresponds to the trivial or topological bandgap frequency range measured from the bulk. In the trivial case, there are no transmission peaks inside the bandgap, consequently no edge modes are measured on the interface between free space and the trivial sample. This is different in the topological case. Indeed, transmission peaks appear for frequencies at the beginning of the bandgap. Looking at the field map corresponding to those peaks, one can see a mode propagating along the edge of the topological sample, and turning around the first corner of the triangle. Since free-space acts as a trivial insulator for such a subwavelength varying field, the edge of the topological sample is considered as an interface between a trivial system and a topological one whose band structure is inverted. This gives birth to edge modes propagating along this domain wall, which we observe experimentally. Note, however, that contrary to the edge modes between two bulk domains, these edge modes are not robust: this is due to the very large breaking of C6 symmetry between the crystal and air. Indeed, it is impossible to define two degenerate pseudo-spin states that are compatible with C6 symmetry in a continuous medium such as air.

**Experimental study of the interface between the two samples.** We present a study of the interface experiment complementary to what is shown in the main text (Fig. 4). The results, displayed on Supplementary Fig. 4, focus on the other parts of the spectrum that is the frequencies that do not belong to the intersection of the bandgaps of the two samples. For frequencies below 4.85 GHz, none of the samples' band structure has a bandgap, therefore propagating modes are excited in both metamaterials. Both sides of the interface lighten, as we can see in the field map (Supplementary Fig. 4b). Between 4.85 GHz and 4.9 GHz, the behavior of each sample is now different. Indeed, while these frequencies correspond to bulk bands of the trivial sample, they fall within the bandgap of the topological one. Moreover, as explained in the previous section, the latter possess propagating modes along its edges with free space at these frequencies. This situation corresponds to the field map of the Supplementary Fig. 4c. For the situation in Supplementary Fig. 4d, the

topological sample does not have edge modes with free space. Finally, in the field map (Supplementary Fig. 4e), alike what happens in (Supplementary Fig. 4b), corresponds to the excitation of bulk bands in both samples.

**Numerical simulations of the interface between the two samples.** We carried out the equivalent of the interface experiment using our semi-analytical model (Supplementary Fig. 5). The spectrum is calculated at one point in the middle of the interface. As for the experiment, peaks appear inside the total calculated bandgap (blue shaded area on the spectrum). In the corresponding calculated field maps, displayed in (Supplementary Fig. 5b,c, we recognize the two modes guided along the interface between the two samples (Fig. 4c,d of the main text). Moreover, we carry out other computations simulating a longer domain wall between the exact same two systems, and adding a crystalline perfectly matched layer built by adiabatically adding absorption losses to the dipoles forming the last ten crystal columns at the right of the figure. This allows us to guarantee no reflection at the end of the interface, and look at the symmetry of a purely forward propagating interface mode. The field maps corresponding to the two guided modes are displayed in Supplementary Fig. 5d (low-frequency) and Supplementary Fig. 5e (high-frequency). They are exactly the same as the ones simulated in the case of the experimental samples. This is a proof that the modes we measure experimentally at the interface between two media made of quarter wavelength resonators also exist when considering analytical resonant point scatterers, thus confirming the generality of our approach. In addition, the PML simulation confirms the good matching of the edge modes in Supplementary Fig. 5b,c, which are radiated at the end of the interface.

**Numerical study of the robustness of the guided modes.** We carry out another set of numerical simulations so as to account for the relative robustness of the guided mode along a tortuous interface between a trivial and a topological medium. Hence, we impose a series of bends to the guide as displayed on Supplementary Fig. 6. Each field map presented here is calculated for a feeding frequency that corresponds to the center of the Brillouin zone of the interface, meaning that all cells are in phase. The first simulation (Supplementary Fig. 6a for the procedure and Supplementary Fig. 6d for the results) prove that our subwavelength guided mode is efficiently propagating along a curved interface involving turns and changes in direction. We implement another complex path for the subwavelength guided mode: a sharply designed cavity shaped as the Eiffel tower (Supplementary Fig. 6b,e). The source is placed at the bottom of the tower's first floor. The field is guided from the left side (blue arrow) and from the right side (green arrow) of the source and then turn around the complex contour of the tower, exciting the cavity. This way we build a complex cavity where the electric field is confined on a subwavelength scale. To study the coupling of this guided mode to the radiation continuum, we remove the crystal at the top of the previous Eiffel tower simulation (Supplementary Fig. 6c,f). Unlike before, once the electric field has reached the top of the tower, it does not turn around but leaks out with the right rate, due to good matching with the environment. Hence, subwavelength edge states can be coupled to the far field. Again, one can notice the difference between the wavelength outside and inside the broadcasting tower. To conclude, this numerical study shows that this system allows to robustly guide waves around sharp corners at a subwavelength scale, and interact with other systems placed in the far field.

**Data availability.** The data that support the findings of this study are available from the corresponding author upon reasonable request.

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

## Acknowledgements

The research leading to these results has received funding from the People Program (Marie Curie Actions) of the European Union's Seventh Framework Program (FP7/2007–2013) under REA grant agreement n. PCOFUND-GA-2013-609102, through the PRESTIGE program coordinated by Campus France. This work is also supported by LABEX WIFI (Laboratory of Excellence within the French Program 'Investments for the Future') under references ANR-10-LABX-24 and ANR-10-IDEX-0001-02 PSL* and by French National Research Agency under reference ANR-13-JS09-0001-01. S.Y. acknowledges funding from French Direction Générale de l'Armement.

## Author contributions

S.Y. performed the experiment. R.F. performed the numerical and semi-analytical simulations. T.B. developed the sample fabrication procedure. F.L. and G.L. supervised the project. All authors contributed to the research work and participated to the redaction of the manuscript.

## Additional information

**Competing interests:** The authors declare no competing financial interests.

