## [Peer Review File · Nature Communications]

Reviewers' comments:

Reviewer #1 (Remarks to the Author):

The authors present a very interesting study of topological wave propagation that combines both modelling and experiments. This is a timely topic and the authors provide a novel and general approach that relies on patterning the deep subwavelength resonant elements of suitably designed metamaterials. It is a solid and elegant piece of work that in my opinion deserves to be published in Nature Communications.

I have a list of comments that I would like the authors to address before recommending publications.

Abstract: The authors do a good job at conveying their excitement to a general audience. I am wondering, however, if it is necessary to use the words "amazing" (twice), "exciting" (once) and "intriguing" (twice) just in the abstract. To some extent the concluding sentence seem also a bit over the top. As I stated above this is a solid and scholarly piece work that does not need these embellishments. Such expressions may come across also as rather subjective and a bit slogan like.

p.4 : Can you clarify why it is not possible to describe the metamaterial simply in terms of effective permittivity? what are the requirements to be able to do so? This is a crucial point that needs to be emphasized.

p.4 : Multiple scattering methods are commonly used in electronic condensed matter to compute band structures (e.g. Korringa-Kohn-Rostoker Green function methods). Hence, I would like the authors to explain why they expect a striking difference between electronic systems and the one they consider. In particular, do they expect that a tight-binding model should NOT be able to describe their crystals? If so, that is extremely interesting (and should be discussed). But if that is really the case, what is the basis for the analogy between resonant elements and atoms in crystals, mentioned first in the introduction, and then expanded all along the manuscript?

from p.5 : It seems to me that the "mathematical folding" due to a supercell is NOT A PHYSICAL effect as long as the "supercell" has no physical existence. In particular, most of the crossing points of figure 1d are consequences of a chosen BZ with no real physical meaning. ONLY when the hexagons are deformed, the supercell acquire a physical meaning. Indeed, it is useful to have the picture 1d to easily compare the band structure with e.g. 2e and 2f, but it should be underlined that this band structure is not really physical.

p.5 : A time-reversal squaring to +1 does not always prevent topological properties to be protected by TRI. This is indeed true in $d < 4$ for TRI alone, but the general statement is not correct.

p.5 : A challenge (which one) is "addressed by augmenting time-reversal symmetry with another symmetry operation". This should be explained more. The authors probably mean to say that they design an "effective/composite fermionic time-reversal" defined as the composition of a crystal symmetry and the bosonic TR operation, and use the same formalism as with fermionic TRI system for this composite crystalline symmetry, which works as long as the additional symmetry is preserved.

p.6 : on figure 2

- in 2c the hexagon should probably be rotated to match the orientation of figure 2a and allow an easy comparison with 2d
- it may help to put the labels s, p, d, f on the images in fig. 2c and 2d

- the gray vs black bands of fig. 2e and 2f make them hard to see well, the authors may want to change something (colors? dashed lines? thicker lines?)

p.7 (and next): I wonder whether there is one canonically trivial phase in this system, or various topologically distinct phases. On the one hand, the band inversion is an argument in favor of the existence of a canonically trivial phase, but on the other hand the presence of crystal symmetries in the composite TRI can give rise to unintuitive results. For instance, I wonder if a change in the boundary conditions with the vacuum/air (i.e., how the crystal is terminated) could also exchange the "trivial" and "topological" phases (as in the SSH model). Can the authors comment on this issue?

p.7 : Please explain what "metamolecules" means here.

p.7 : The geometric explanation is interesting, but not completely obvious to me ... I would like to have more details on that.

p.7 : is "more energetic" correct? the frequency is higher, but what is the energy metric here?

p.7 : is the 4-fold degeneracy ever real? (see above)

p.7 : what does "the top and bottom bands remain doubly degenerate at Gamma, in accord with the two IR of the C6 point group" mean?

There are more than two IR in C6.

p.8 : The central paragraph in page 8 is supplemented with expressions like "directly follows", "indeed", "clearly evident". For the convenience of the reader, is it possible to replace them with explanations of the facts deemed to be obvious?

p.8 : why are p_1 and p_2 (resp d) used?

p.9 : in figure 3, is the "normalized amplitude" in linear scale (as opposed to log scale)? why? what are the uncertainties and resolutions in the measurements?

The patterns of figures 3c and 3d are convincing, but comments on the deviations from an "ideal" behavior (e.g. why are there p-type cells on the boundary of the lattice in the s mode of 3c?) would be welcome. In addition, only several peaks in the amplitude vs frequency plots are labeled, what are the others?

p.9 : it is not clear exactly how the system is excited (globally or locally? at a given frequency? and what is the bandwidth of the excitation? how are the fields normalized, if so? and so on) ... however, this information is important to understand the results, and should be explained in the manuscript (maybe not with all technical details, which belong in the Methods, but what is measured should be explained)

p.10 : what does "a large area" for the spatial average mean? what is this area exactly? the sample is quite small and boundary effects are expected ...

p.10 : a band inversion is NOT equivalent to a topological phase! It is indeed an important CLUE, but not a proof ... in this case, the match with the theory gives a strong evidence that there is a topological phenomenon in the experimental system, but again, it is not a direct experimental proof (experimentally measuring a topological invariant would be).

p.10 : the authors say they have an "excellent quantitative agreement" between finite-element simulations results and experimental results. However, there are no QUANTITATIVE comparison of both (only qualitative ones). I agree that quantitative comparison (with numbers, uncertainties, ...) would be of great interest, and encourage the authors to do it.

p.11 : figure 4 has a somehow misleading title as the "topological edge modes" are not really topological due to the breaking of the C_6 symmetry at the interface, as the authors discuss later

p.11 : It would be useful to write "experiment" and "theory/numerics/..." on the images of fig. 3c. Is the normalization of the fields in theory and experiments the same?

p.11 : the experimental data is convincing. However, I would like to see transmission data through an edge state, e.g. in presence and in absence of a defect, to really be convinced of their "robustness".

p.11 : again, what is the bandwidth of the excitation? In particular, how does it compare to the size of the small gap of fig. 3d ? it is probably smaller as the gap is resolved, but to what extent?

p.13 : the authors attribute the robustness to defects of edge states to their symmetry protection. Yet, the corresponding symmetry is broken. Can the authors comment?

p. 13 : I do not see why the fact that the field amplitude is the same before and after the defects is a conclusive evidence, at least as long as how the system is excited is not made precise (e.g. is the transmission "through" the defect measurement)

p. 13 : "tested it for acoustic metamaterials" : experimentally?

Methods : It would be good to have more details.

For example, what is exactly the "homemade antenna"?

What is the semi-analytical model?

Reviewer #2 (Remarks to the Author):

In this work, the authors demonstrated an analogue of quantum spin Hall like system with deep subwavelength metamaterials. Instead of the usual description of metamaterial with effective parameters, here the authors show that the lattice structure is quite important. The experimental demonstrations are good and clear. The manuscript is well-written and easy to follow. However, I think there are still some issues (listed below) need to be addressed before this work is accepted to be published in Nature Communication.

Major issue:

The one-way property demonstrated in Fig. 4f. Due to the breaking of C_6v symmetry on the boundary, the two "spins" of the surface states hybridize and open a minor gap. This is the reason why there is a gap between the HF and LF state in Fig. 4d. Due to the hybridization, the surface state at these two frequencies (HF and LF) no longer exhibit one-way property. For frequency below LF or above HF, the one-way property should still be ok. Hence I think the authors should double check their results for Fig. 4f.

I think the credit of Ref. [10] is not given properly, or more precisely, not enough. The interpretation in this manuscript is quite clear, but in my understanding, the main idea of such construction is in Ref. [10]. However, this relation is not pointed out explicitly, instead, Ref. [10] is only cited on page 1 and 5 with other works. Or if this work's idea is quite different from Ref. [10], what is the major difference?

There is also another similar work on arXiv. Instead of metamaterial in deep subwavelength region, that system focuses on photonic crystals. Yuting Yang et al, arXiv:1610.07780. This highly related work should at least be cited.

When introducing topological metamaterial, I think the authors should cite the following works. W. Gao et al, PRL 114, 037402 (2015), realize topological photonic phase in chiral hyperbolic metamaterials

M. Xiao et al, PRL 117, 057401 (2016), realize topological semimetal in chiral hyperbolic metamaterials

Also M. G. Silveirinha's works, who contributes a lot in topological metamaterials.

Fig. 3c, 3d. Is there a reason why the frequencies of the s states are so different in these two figures? According to Fig. 2f, the s state in Fig. 3d is no longer a s state, the band already folds back.

The frequencies of the band gap in Fig. 4b and Fig. 4d are different. What is the main reason for this frequency shift? And how do the authors decide the nontrivial band gap region in Fig. 4b?

Minor issue:

By the end of the introduction paragraph, the authors mentioned that "It certifies that metamaterial crystals can be engineered whose potential greatly surpass that of conventional ones". This statement is vague or at least needs further interpretation.

By the end of the first paragraph on page 4: "the first consequence of the crystalline nature of the metamaterial appears: the dispersion relation which links the frequency to a wavenumber is direction dependent." While this sentence by itself is not wrong. However, this sentence is not consistent with the following sentence "mostly gives rise to isotropic properties"

In the second paragraph on page 4, the authors change from triangular lattices to honeycomb lattices and then obtain Dirac cones. When I was reading this paragraph, I feel a little misleading as the triangular lattice also supports Dirac cones, but at a higher frequency. I understand that the authors are working in deep subwavelength region. However, I think it is better to point out this explicitly.

The first sentence on page 5. "Going from the triangular lattice to the honeycomb one can be pictured as a folding of the band structure, the area of the first Brillouin zone being divided by 3." I guess here is a typo, from triangular lattice to honeycomb lattice, the first BZ is divided by 2 not 3.

Two sentences below. "Thus, we consider the supercell containing a full hexagon of resonators instead of the primitive one containing only the 2 resonators (Fig. 1d)". Here the citation of figures are not proper. I think it should be "Thus, we consider the supercell containing a full hexagon of resonators (Fig. 1d) instead of the primitive one containing only the 2 resonators (Fig. 1c)"

In the middle of the first paragraph on page 12. "In stark contrast, leaky wave out of plane radiation is not allowed along the interface due to the subwavelength nature of the edge mode." I get a little confused here. In the demonstration, the edge mode and nontrivial band gap is inside the light cone, hence I think it can couple with the light cone. Do I miss something here?

Reviewer #1: *The authors present a very interesting study of topological wave propagation that combines both modelling and experiments. This is a timely topic and the authors provide a novel and general approach that relies on patterning the deep subwavelength resonant elements of suitably designed metamaterials. It is a solid and elegant piece of work that in my opinion deserves to be published in Nature Communications. I have a list of comments that I would like the authors to address before recommending publications.*

Authors: We would like to thank the referee for his/her careful study of our manuscript and for his/her positive opinion and recommendation. We address in the following all his/her comments.

R1: *Abstract: The authors do a good job at conveying their excitement to a general audience. I am wondering, however, if it is necessary to use the words “amazing” (twice), “exciting” (once) and “intriguing” (twice) just in the abstract. To some extent the concluding sentence seem also a bit over the top. As I stated above this is a solid and scholarly piece work that does not need these embellishments. Such expressions may come across also as rather subjective and a bit slogan like.*

A: Looking back at our abstract, we agree with the reviewer and have limited the use of hyperbolic expressions. The final sentence has been toned down.

R1: *p.4 : Can you clarify why it is not possible to describe the metamaterial simply in terms of effective permittivity? what are the requirements to be able to do so? This is a crucial point that needs to be emphasized.*

A: We thank the reviewer for this relevant comment. The most basic homogenization theories (used for instance to describe dielectrics) assume that spatial dispersion is negligible (i.e. the polarization state at one point cannot depend on the field incident on another point), which is equivalent to saying that the effective parameters cannot depend on the wave number k . This assumption is clearly violated in a periodic structure if multiple scattering occurs, since the incident field on a scatterer will polarize the other scatterers. In metamaterials with weak spatial dispersion, this issue is usually overcome by increasing the number of effective parameters that are needed to describe the material, taking into account both electric and magnetic polarizabilities, and magneto-electric coupling. (see for instance PRB 84, 075153 (2011)). Yet, even this complicated description is not enough to capture the physics of wave propagation in a subwavelength arrangement of resonators, where multiple scattering is very strong (as we have shown in a previous work, Nature 525, 77-81, 2015). In such a case, only a crystal (Bloch) description can effectively capture the physics of the material. We have made this point clearer in the revised manuscript and added a reference on the limits of homogenization methods in the presence of strong spatial dispersion.

R1: *p.4 : Multiple scattering methods are commonly used in electronic condensed matter to compute band structures (e.g. Korringa-Kohn-Rostoker Green function methods). Hence, I would like the authors to explain why they expect a striking difference between electronic systems and the one they consider. In particular, do they expect that a tight-binding model should NOT be able to describe their crystals? If so, that is extremely interesting (and should be discussed).*

A: This is a very interesting remark. In general, the tight binding model does not directly apply to subwavelength arrangements of resonant scatterers. We expect striking differences since the coupling between resonators is of propagative nature and extended well beyond nearest neighbors, and not evanescent. This can be seen in Fig. 1c in the case of a honeycomb lattice, whose first branch features a polariton-like dispersion, very different from the sinusoidal profile expected from the TB model. We have clarified this distinction in the revised manuscript.

R1: *But if that is really the case, what is the basis for the analogy between resonant elements and atoms in crystals, mentioned first in the introduction, and then expanded all along the manuscript?*

A: This is an important point and we are happy to clarify it. When we mention the analogy between metamaterial inclusion and atoms in crystals, we do not refer to an analogy between electromagnetic and electronic propagation, but we talk about the analogy between *electromagnetic propagation* in natural materials, and *electromagnetic propagation* in metamaterials. In particular, in both cases the subwavelength scatterers (in natural crystals, atoms that can be polarized) and metamolecules (artificial resonant structures, that can be polarized) interact with an external wave to define the electromagnetic properties of the material. In both cases, the *structure* and the *composition* of the material are important. This is the basis of the analogy that is made throughout the manuscript, between artificial and natural materials. In the revised manuscript, we avoid making direct analogies with electronic systems, since such an analogy is indeed disputable, as the reviewer points out.

R1: *from p.5 : It seems to me that the "mathematical folding" due to a supercell is NOT A PHYSICAL effect as long as the "supercell" has no physical existence. In particular, most of the crossing points of figure 1d are consequences of a chosen BZ with no real physical meaning. ONLY when the hexagons are deformed, the supercell acquire a physical meaning. Indeed, it is useful to have the picture 1d to easily compare the band structure with e.g. 2e and 2f, but it should be underlined that this band structure is not really physical.*

A: We entirely agree with the reviewer when he/she says that the folding we operate on the hexagonal lattice is not physical. The supercell effectively becomes a real description of the crystalline metamaterial only when the hexagons are shrunk or expanded. That is what we meant by “mathematical folding” in opposition with a “physical” one. However, this being not clear enough we took the reviewer’s remark into account in the revised manuscript and explained this point better.

R1: *p.5 : A time-reversal squaring to +1 does not always prevent topological properties to be protected by TRI. This is indeed true in $d < 4$ for TRI alone, but the general statement is not correct.*

A: We thank the reviewer for this constructive remark. In our bi-dimensional study the fact that $T^2 = +1$ is indeed an obstacle to tailor topological properties. However, we agree that it cannot be generalized for all dimensions, and we included the remark in the revised version of the manuscript.

R1: *p.5 : A challenge (which one) is "addressed by augmenting time-reversal symmetry with another symmetry operation". This should be explained more. The authors probably mean to say that they*

design an "effective/composite fermionic time-reversal" defined as the composition of a crystal symmetry and the bosonic TR operation, and use the same formalism as with fermionic TRI system for this composite crystalline symmetry, which works as long as the additional symmetry is preserved.

A: Again, we thank the referee for this helpful remark and opportunity to improve the clarity of our work. It clearly shows that he/she has perfectly understood the procedure we use in order to describe our classical system with a fermionic formalism. We understand that this point should be more precise in the manuscript for the sake of the reader's comprehension. Therefore, we follow his/her advice and develop this point further in the revised version.

R1: *p.6 : on figure 2*

- in 2c the hexagon should probably be rotated to match the orientation of figure 2a and allow an easy comparison with 2d

A: Good point. We have rotated the view in Fig. 2a to implement the referee's suggestion.

R1: *- it may help to put the labels s, p, d, f on the images in fig. 2c and 2d*

A: We thank the reviewer for paying attention the figure labels. The modes symmetries s, p, d and f are related to hexagons, since defined for a C₆-symmetric object. However, in 2d, the metamolecule no longer has C₆ symmetry, and the label is not rigorously justified. We preferred to keep the labeling as it is in the figure, and talk about these modes as being "monopolar", "dipolar", "quadrupolar", and "hexapolar" in the text, which is a correct way of describing them.

R1: *- the gray vs black bands of fig. 2e and 2f make them hard to see well, the authors may want to change something (colors? dashed lines? thicker lines?)*

A: We thank the reviewer for this remark. The lines have been made thicker to make them easier to read following the comment.

R1: *p.7 (and next): I wonder whether there is one canonically trivial phase in this system, or various topologically distinct phases. On the one hand, the band inversion is an argument in favor of the existence of a canonically trivial phase, but on the other hand the presence of crystal symmetries in the composite TRI can give rise to unintuitive results. For instance, I wonder if a change in the boundary conditions with the vacuum/air (i.e., how the crystal is terminated) could also exchange the "trivial" and "topological" phases (as in the SSH model). Can the authors comment on this issue?*

A: This is a very interesting comment, and we are glad to provide an answer. In fact, there can be no such exchange between the topological and trivial phase in our system. Indeed, we have calculated numerically the spin Chern numbers of the two bands of interest, finding non-zero values, and demonstrating unambiguously the inherent topological distinction between the two phases. *This calculation does not depend on a particular crystal termination.*

R1: p.7 : *Please explain what "metamolecules" means here.*

A: Here the “metamolecules” are the building blocks of the crystalline metamaterial. As presented in Fig. 1b, wires can be seen as a “meta-atoms” whose resonance frequencies are analogs of the energy levels of an actual atom. In the case of the topological and trivial lattices, the building blocks are no more single resonators but clusters of six of them. This way we design “metamolecules” from these “meta-atoms”. In the same way, these metamolecules have resonant modes, represented on Fig. 2c,d which are macroscopic analogs of the atomic orbitals of the corresponding molecules. We explain this point better in the revised version of the manuscript.

R1: p.7 : *The geometric explanation is interesting, but not completely obvious to me ... I would like to have more details on that.*

A: We thank the reviewer for his/her interest in this particular explanation, and we are keen on providing more details.

First, it is useful to notice that wave propagation among a locally resonant metamaterial structured as a triangular lattice of resonators is described by a polariton as represented in Fig.1b. This behavior arises from the hybridization between the propagating wave and the resonant mode of the wires (dipolar along the wire yet monopolar transversally to it), previously referred as “meta-atoms”. So, we can understand the band as being due to the mode of each resonator hybridizing with the continuum of planes waves.

We can describe the behavior of the topological and trivial crystals in a very similar way, but we have to take into account the fact that the meta-atom is now a meta –molecule, composed of 6 wires (and the metamolecules of the trivial and topological systems are different, as represented in Fig. 2c,d). Now, we no longer have a single resonance, but six different resonant modes. Each of these 6 modes will hybridize with the propagative modes of the background to create a band. At the gamma point, these six bands will appear in the same order (along the frequency axis) than the six modes of the metamolecules.

This simple description is enough to explain the band inversion. Since the lattice is C_6 symmetric, we know that any C_6 symmetric unit cell will support s, p, d and f modes, but we don't know their order of appearance (along the frequency axis). To determine this order, we look at the modes of the metamolecules alone (Fig.2c,d) and, reporting these distributions in the crystal, we can determine whether they induce a s,p,d, or f mode distribution on the shrunk or expanded hexagons in a unit cell (This is done in the inserts of Fig. 2e,f).

In the case of the trivial sample, the metamolecule being already a hexagon, the hybridized crystalline modes are directly identified as s, p, d and f when you increase the frequency. Similarly, for the topological crystal, one needs to look at the modes of a metamolecule and determine whether they induce s, p, d or f modes on the expanded hexagons. Once this is done (insert of Fig. 2f), we notice that the frequency of the p and d modes defined on the expanded hexagons are inverted compared to the ones of the shrunk hexagons in trivial sample. This simple intuitive picture explains the topological band inversion occurring in these polaritonic metamaterial.

In order to ensure the reader's comprehension of this explanation, we detail it more precisely in the revised manuscript. Also, Figure 2 has been modified to make this explanation clearer.

R1: *p.7 : is "more energetic" correct? the frequency is higher, but what is the energy metric here?*

A: We thank the reviewer for asking this question. Yes, it is correct to say that higher order subwavelength modes are more energetic (energy = frequency if we compare these modes to electronic orbitals). Since they contain more spatial variations on a subwavelength scale, higher order modes resonate at higher frequencies. We have clarified what we mean in the main text.

R1: *p.7 : is the 4-fold degeneracy ever real? (see above)*

A: No, it is never real, as we explain above in our response. We could have presented directly the gapped band structures, but we prefer explaining step by step how the design is found for the sake of clarity. The four-fold degeneracy is simply the starting point of our reasoning. This is clarified in the revised manuscript.

R1: *p.7 : what does "the top and bottom bands remain doubly degenerate at Gamma, in accord with the two IR of the C6 point group" mean? There are more than two IR in C6.*

A: What we meant is that at the Gamma point there are two two-dimensional representation in the C6 symmetry point group associated with the odd and even parities with respect to spatial inversion. Thus we have one s mode, two degenerate p modes, two degenerate d modes, and one f mode. To make things clear to a wider range of readers, in the revised manuscript we just say that the degeneracy of p and d modes at Gamma is guaranteed by C6, and cite a work on the topic [PRL 114, 223901, 2015].

R1: *p.8 : The central paragraph in page 8 is supplemented with expressions like "directly follows", "indeed", "clearly evident". For the convenience of the reader, is it possible to replace them with explanations of the facts deemed to be obvious?*

A: We have modified this paragraph to improve the description of the figure. We are grateful to the reviewer for his/her suggestions to further improve the readability of our manuscript.

R1: *p.8 : why are p_1 and p_2 (resp d_1 and d_2) used?*

A: We thank the reviewer for asking this question. We show in the manuscript that applying a small deformation to the hexagons in order to create a trivial or a topological phase results in the breaking of the mathematical four-fold degeneracy in the center of the Brillouin zone. In each case this opens a bandgap surrounded by two pairs of two-fold degenerate bands related to the modes p_1 and p_2 on the one hand and d_1 and d_2 on the other hand. Following Ref [10], we emulate from these two pairs of degenerate modes the two pseudo-spins in the topological and trivial crystalline metamaterials, namely the modes $(p_1 \pm ip_2)/\sqrt{2}$ and $(d_1 \pm id_2)/\sqrt{2}$. They correspond to positive and negative angular momentum of the out-of-plane electric field. This is done simply to highlight better the analogy with the quantum spin-Hall effect by defining the spinning states. Taking into account the reviewer's question, we clarify this point on the revised version of the manuscript.

R1: p.9 : in figure 3, is the "normalized amplitude" in linear scale (as opposed to log scale)? why? what are the uncertainties and resolutions in the measurements? The patterns of figures 3c and 3d are convincing, but comments on the deviations from an "ideal" behavior (e.g. why are there p-type cells on the boundary of the lattice in the s mode of 3c?) would be welcome. In addition, only several peaks in the amplitude vs frequency plots are labeled, what are the others?

A: We thank the reviewer for these constructive comments about the experimental results. In Fig. 3 the spectra are indeed displayed in linear scale because we think it is a clearer way to account for the several experimental transmission bands and bandgaps. For instance, we see right away the band where the field goes to zero, and the corresponding discussion is made clearer.

In addition, we have added some data about the precision of our measurements in the Methods.

Moreover, we understand that the ideal behavior the reviewer is mentioning would occur for the case of an infinite medium, with no boundaries. One solution would be to make larger samples to diminish the influence of the boundaries on the mode symmetries. However, the modes presented in the insets of the Fig. 3c are taken on hexagons inside the bulk and not directly on the boundaries and clearly they are a strong proof of the mode symmetries in the crystalline metamaterial at these frequencies. Furthermore, because of the losses inherent to the metallic wires, the peaks we measure have a finite bandwidth which may lead to the overlapping of the corresponding modes. This explains why these experimental modes, although showing a very good agreement with the simulated ones, are not perfectly identical to them. We comment on that deviation in the revised version of the manuscript.

Finally, as the reviewer has noticed, we observe several peaks in the experimental spectra. Their presence is due to the finite size of the sample. Compared to the continuum of modes we obtain when we compute the band structure of the corresponding infinite crystal, the fact that we have a finite number of resonators in our sample discretizes the spectra. Therefore, it is perfectly normal to measure several peaks instead of continuous bands of transmission. We could have avoided that by using a much larger sample, but our fabrication method currently limits us to 10cm or so wide samples. These points are now more detailed in the revised manuscript.

R1: p.9 : it is not clear exactly how the system is excited (globally or locally? at a given frequency? and what is the bandwidth of the excitation? how are the fields normalized, if so? and so on) ... however, this information is important to understand the results, and should be explained in the manuscript (maybe not with all technical details, which belong in the Methods, but what is measured should be explained)

A: We are thankful to the reviewer for this relevant suggestion. We probe the medium in its close vicinity (see Methods) with a network analyzer. This allows us to make a broadband measurement (1 GHz) in one step, whose peak bandwidth is 625 kHz. It is by far sufficient to resolve any physical peak of the sample. Indeed, losses inherent to the metallic wires impose a lifetime below the microsecond, which is smaller than the 1.6 μ s corresponding to the peak resolution we have. Following the reviewer's advice, we have given more details in the main text about the experimental setup, including about the excitation.

R1: *p.10 : what does "a large area" for the spatial average mean? what is this area exactly? the sample is quite small and boundary effects are expected ...*

A: We are happy to give more details on the averaging of the spectra. It is true that the sample is small compared to the wavelength of the incoming wave however it is possible to define an area inside the bulk which covers six unit cells in the center of the samples. We make the averaging on this area and it give good averaging of the measured spectra. Indeed, the bandgaps and transmission peaks occur for frequency ranges that are consistent with the simulation. We added more details in the methods.

Moreover, we agree with the referee when he/she expects the boundaries to affect the spectra. It is particularly the case for the topological sample (Fig. 3d) where smaller transmission peaks are found at the beginning of the bandgap (blue shaded area), at 4.89 GHz. They correspond to modes propagating along the edges of the sample, at the boundary with air. Their study is now more detailed in the Methods.

R1: *p.10 : a band inversion is NOT equivalent to a topological phase! It is indeed an important CLUE, but not a proof ... in this case, the match with the theory gives a strong evidence that there is a topological phenomenon in the experimental system, but again, it is not a direct experimental proof (experimentally measuring a topological invariant would be).*

A: We are in entire agreement with the referee on the fact that the band inversion is not a sufficient condition for the occurrence of a topological phase. That is why we also computed relevant invariants (spin-Chern numbers) corresponding to our system to ensure that we actually study topological phases in our work. However, we understand that there was an ambiguous sentence in the original manuscript and we modified it in the revised version of the manuscript.

R1: *p.10 : the authors say they have an "excellent quantitative agreement" between finite-element simulations results and experimental results. However, there are no QUANTITATIVE comparison of both (only qualitative ones). I agree that quantitative comparison (with numbers, uncertainties, ...) would be of great interest, and encourage the authors to do it.*

A: We have fixed the corresponding sentence to explain where the agreement is quantitative, and where it is only qualitative. In short, there is excellent quantitative agreement with the frequency computed for each bands via FEM simulations and the frequency range where the corresponding modes are observed experimentally. The shapes of the modes show qualitative agreement.

R1: *p.11 : figure 4 has a somehow misleading title as the "topological edge modes" are not really topological due to the breaking of the C₆ symmetry at the interface, as the authors discuss later.*

A: We believe it is correct to say that the edge modes are topological since they exist because of the difference in the topology of the two surrounding bulks crystals. The claim that cannot be made is that they are *topologically protected*, because of C₆ symmetry breaking at the interface. This is similar to

the case of a quantum spin-Hall system in the presence of external magnetic field: the edge currents are topological but not topologically protected. Our simulations and measurement demonstrate, however, that these modes are still extremely robust and show negligible backscattering in the presence of turns and defects. We hope that the referee grants us that our phrasing is, strictly speaking, correct.

R1: *p.11 : It would be useful to write "experiment" and "theory/numerics/..." on the images of fig. 3c. Is the normalization of the fields in theory and experiments the same?*

A: Figure 3 is purely experimental. There are no numerical data in this figure. The numerical predictions associated with this figure are presented in the methods section. Since the experimental data is obtained from a measure of the evanescent field on top of the wire, and the numerical predictions are obtained from a simple two-dimensional model, the excited amplitudes in the numerical simulation and in the experiment are not expected to be of the same magnitude. Therefore, the plots are not normalized to the same value. We stress that our goal with the numerical model is not to reproduce exactly the field excited in the experimental sample, but to capture the essential physics of topological polaritons with a simple model, including the material dispersion and mode symmetries.

R1: *p.11 : the experimental data is convincing. However, I would like to see transmission data through an edge state, e.g. in presence and in absence of a defect, to really be convinced of their "robustness".*

A: As the reviewer notes, our field map measurements in presence and absence of defect clearly demonstrate that it is possible for the electromagnetic energy to be transmitted past a PEC wall put right along the route of the mode. As we further discuss below, this is an *absolutely unique property*, extremely difficult to achieve at such a deep subwavelength scale. This would never happen in a topologically trivial subwavelength waveguide, especially for such a confined wave.

To the best of our knowledge, there is only one other solution to guide waves at the subwavelength scale: building a waveguide based on evanescently coupled defects (see arxiv:1604.08117) placed in a bandgap medium. In these waveguides, a PEC wall would completely cut the evanescent coupling between defects and lead to total reflection and no transmission. In stark contrast, our field map shows that the *field excited after the PEC wall has the same amplitude as the field excited before*, confirming the ability of the topological waveguide to work around the obstacle and transmit a large portion of the incident power through the defect. We believe that this is the first direct experimental observation of such a level of robustness for a deeply confined subwavelength electromagnetic field.

The reviewer is also asking for a quantitative comparison of the transmission with and without the defect. To perform such a measurement, one would need to achieve a perfect calibration of the antenna in order to guarantee that the matching of the antenna to the waveguide is the same in absence or presence of the PEC wall. Only in these conditions one could reasonably compare the amplitude levels found at the back of the wall. Unfortunately, our broadband excitation method prevents us from

matching the probe to the sample over a wide range of frequencies, and our magnetic loop antenna leaves us with no simple degree of freedom to tune the matching of this system. Because of these issues, we cannot provide the reviewer with a meaningful comparison of the transmission data with and without the defect. We are happy to provide a curve in this response, which shows reasonably equivalent measures of the transmission with and without the defect, especially for the HF peak. Yet since in neither case the antennas are matched to the waveguide, we do not want to include it in the manuscript. Indeed, it could very well overestimate the robustness of the waveguide, for instance if the emitting antenna happens to be more matched to the waveguide with defect: this would be a misleading information.

That being said, as the reviewer remarks, the data provided in the field maps is sufficient enough to support the claims made in the paper, and all these refinements are clearly beyond the scope of the present work. Here, we focus on demonstrating the concept of topological surface polaritons, and provide the first experimental proof of a large degree of robustness at the subwavelength scale, which is a remarkable result for such confined fields.

R1: *p.11 : again, what is the bandwidth of the excitation? In particular, how does it compare to the size of the small gap of fig. 3d ? it is probably smaller as the gap is resolved, but to what extent?*

A: This is indeed an important precision, and we now provide it in the methods. The bandwidth of excitation is 625 kHz, corresponding to 0.3% of the size of the small gap. We included this important information in the methods.

R1: *p.13 : the authors attribute the robustness to defects of edge states to their symmetry protection. Yet, the corresponding symmetry is broken. Can the authors comment?*

A: We are grateful to the referee for asking this well-founded question. The level of robustness turns out to be dependent on the way the interface breaks the C_6 symmetry. If breaking is weak, that is if

the hexagons in the topological sample are not too different from the ones in the trivial sample, the modes propagate along the boundary and take sharp corners as we have shown it numerically in the Methods (Extended data Fig. 6). If the C_6 symmetry breaking is stronger, as it is the case in an interface between the topological sample and the air, the robustness of the mode is hampered as it is shown in the Methods (Extended data Fig. 3d). This point is clarified in the revised manuscript. Moreover there are some ways to circumvent this issue: first, instead of an abrupt boundary, which is the worst-case scenario, one can do an “adiabatic” deformation between the two samples across their boundary and tune the desired level of robustness of the mode, at the expense of mode localization. In addition, it has been shown recently that for some specifically designed interfaces the edge modes are not gapped and seem to be perfectly protected. These refinements are beyond the scope of this work and are currently under investigation.

R1: *p. 13 : I do not see why the fact that the field amplitude is the same before and after the defects is a conclusive evidence, at least as long as how the system is excited is not made precise (e.g. is the transmission "through" the defect measurement)*

A: We thank the referee for this relevant comment. We have clarified in the methods how the sample is excited and probed. The sample is indeed excited at the bottom of the sample under the interface, and we probe the electric field in the near field on top of the wires. In these conditions, any electromagnetic energy found after the defect has necessarily been tunneled through the defect. The fact that the field amplitude is the same before and after the defect demonstrates the possibility to have efficient transmission of energy through a defect as drastic as a PEC wall placed right on the path of the wave. As we discuss in further details in a response above (last question on p8), this would be completely impossible in other types of subwavelength waveguides, and already a strong proof that topological polaritons offer unprecedented robustness at the subwavelength scale.

R1: *p. 13 : "tested it for acoustic metamaterials" : experimentally?*

A: We carried out the same type of 3D simulations as those which are presented in the paper but for loss-less subwavelength acoustic Helmholtz resonators displayed on the same subwavelength-scaled lattice. We used the acoustic module of Comsol Multiphysics and obtained the same results as for the wires. We do not yet have an experimental validation in acoustics.

R1: *Methods : It would be good to have more details. For example, what is exactly the "homemade antenna"? What is the semi-analytical model?*

A: We follow the referee’s advice and add more details in the Methods, including a photograph of the antenna we use in our experiment.

Reviewer #2: *In this work, the authors demonstrated an analogue of quantum spin Hall like system with deep subwavelength metamaterials. Instead of the usual description of metamaterial with effective parameters, here the authors show that the lattice structure is quite important. The experimental demonstrations are good and clear. The manuscript is well-written and easy to follow.*

Authors: We would like to thank the reviewer for carefully reading our manuscript and for confirming that our work differs from the usual description of metamaterials. We are also grateful to him/her for confirming that our experiments were well carried out and that the results are presented in a convenient way.

R2: *However, I think there are still some issues (listed below) need to be addressed before this work is accepted to be published in Nature Communication.*

Major issue: The one-way property demonstrated in Fig. 4f. Due to the breaking of C_{6v} symmetry on the boundary, the two “spins” of the surface states hybridize and open a minor gap. This is the reason why there is a gap between the HF and LF state in Fig. 4d. Due to the hybridization, the surface state at these two frequencies (HF and LF) no longer exhibit one-way property. For frequency below LF or above HF, the one-way property should still be ok. Hence I think the authors should double check their results for Fig. 4f.

A: We thank the referee for raising this interesting point. We entirely agree with the fact that breaking the C_6 symmetry on the interface, strictly speaking, prevents *topological protection* of the edge mode. The question is then to determine to what extent this is detrimental to the robustness of the mode. The answer to this question is that the level of residual robustness actually depends on the way the interface breaks C_6 symmetry. For instance, imagine an interface between two domains with very small deformations of the hexagons. Each domain is an insulator, but because the deformations are weak, the gaps are not very large, and consequently the modes are less confined to the interface. At the same time, because the breaking of C_6 symmetry is weak, the minor gap is so small that it has no effect on the properties of these modes, and they remain extremely robust. On the contrary, largely deformed hexagons induce interfaces that support very confined modes, but since C_6 symmetry breaking is larger, the modes are less robust due to spin hybridization. There is thus a tradeoff between mode localization and robustness.

With this in mind, we see that we can play some tricks to circumvent this issue. For instance, we could consider, instead of an abrupt interface, an adiabatic deformation between one sample to the other. Locally, the deformation of the crystal would be so slow that the sample would not feel at all the breaking of C_6 symmetry. The edge modes would be extremely robust. In addition, they would retain their main interesting features: localization to the surface, ultraslow propagation and subwavelength confinement.

Recently, another group went even further and studied (in the different case of crystals) specific interfaces that are able to close the minor gap and restore topological protection (Xia Hu. “Z2 topological photonics derived from crystal symmetry”, presented at Meta16 in Malaga, Spain). These results confirm that the problem of C_6 symmetry breaking at an interface may not be a fundamental limitation of this design.

Experimentally, for the first time, we confirm that these edge modes are still extremely robust even in the presence of a drastic, C_6 incompatible wall defect on their route. In addition, our numerical

simulation demonstrates the robustness of edge modes even if their lateral confinement is also subwavelength. The surprising result of our study is that it really takes a lot to significantly hamper the robustness of these modes: of all the kinds of defects that we have tested, only the interface with air showed noticeable backscattering. We had to interface the crystal with a continuous medium to start destroying the one-way property. This is a clear experimental evidence of the strong robustness of the topological modes.

In the conclusion of the revised manuscript, we have stressed these important points better in order to be clear about the fact that the modes are *topological*, not *topologically protected*, yet still extremely *robust*. We have also added several remarks about the trade-off between localization and robustness, and the possibility to avoid breaking spin-degeneracy using adiabatic interfaces. We hope that with these modifications and our arguments, the referee now does not see this point as limiting for a publication of our work in Nature Communications.

R2: *I think the credit of Ref. [10] is not given properly, or more precisely, not enough. The interpretation in this manuscript is quite clear, but in my understanding, the main idea of such construction is in Ref. [10]. However, this relation is not pointed out explicitly, instead, Ref. [10] is only cited on page 1 and 5 with other works. Or if this work's idea is quite different from Ref. [10], what is the major difference?*

A: We thank the referee for the comment. In the revised manuscript we are happy to give proper credit to this seminal work by explicitly referring to it when we introduce the idea of the C6 pseudo-spin engineering recipe.

The referee's comment also suggests that we may not have conveyed clearly the fundamental differences between the present work and the previous literature. We entirely agree with the fact that the idea of pseudo-spin engineering using C6 symmetry was originated in [10], and applied to non-resonant crystals where the insulating states are based on Bragg interferences, leading to supra-wavelength topological modes (in subsequent works that we also cite).

Conversely, our work demonstrates the possibility to emulate pseudo-spins at the *subwavelength scale*, in crystalline metamaterials where the insulating state and the topology originate from *Fano interferences* and multiple scattering at the subwavelength scale. The transition from [10] to subwavelength crystal is by no means straightforward as the physics are fundamentally different. We have stressed this better in the manuscript by revising the concluding comments accordingly.

R2: *There is also another similar work on arXiv. Instead of metamaterial in deep subwavelength region, that system focuses on photonic crystals. Yuting Yang et al, arXiv:1610.07780. This highly related work should at least be cited.*

A: We are thankful to the reviewer for pointing out this relevant work, and we are happy to take this opportunity to include it in our reference list. Yet it does not in our opinion lower the originality of our proposal since, as noted by the referee, this work demonstrates topological states that are supra-wavelength. Very differently, we demonstrate subwavelength states that are based on different physics (our entire metamaterial crystal is smaller than the wavelength). See the response to the previous comment for more details.

R2: *When introducing topological metamaterial, I think the authors should cite the following works. W. Gao et al, PRL 114, 037402 (2015), realize topological photonic phase in chiral hyperbolic metamaterials*

M. Xiao et al, PRL 117, 057401 (2016), realize topological semimetal in chiral hyperbolic metamaterials

Also M. G. Silveirinha's works, who contributes a lot in topological metamaterials.

A: We thank the reviewer for bringing to our attention these important contributions. We are happy to have the opportunity now to resubmit a manuscript with an updated list of references, and these particular references have been included and appropriately cited.

R2: *Fig. 3c, 3d. Is there a reason why the frequencies of the s states are so different in these two figures? According to Fig. 2f, the s state in Fig. 3d is no longer a s state, the band already folds back.*

A: We thank the reviewer for this relevant question. The difference in the frequency where the maps are taken comes from the fact that the s-band covers a wide range of frequencies, and depending on the sample, even if the location of the source is kept constant, the frequencies that resonantly excite the two finite samples on this branch are different.

Also, note that classification of states as s, p, d, and f states can only be clearly determined at Gamma point. Away from it, the propagating part of the Bloch mode can make them more difficult to recognize. Compared to p, d and f states, s-states are tricky to identify since they only reach the Gamma point at zero frequency, and we cannot access this point experimentally. What we can access is the part of the branch that is already near K point, in the frequency range where the experiment is done. Therefore, there is always a propagating part in these modes that makes it more difficult to identify the mode as being monopolar. The reviewer is correct in pointing out that the state in Fig. 3d has been taken near K point, where the band folds back. This has been chosen because at that frequency, the coupling between our probe and the sample is more efficient than at lower frequencies (we see a larger peak in the spectrum), and we have a good field map. At the same time, it is clear that the field is mostly of monopolar nature, which is the main thing claimed by this figure panel and the only thing that really matters.

In Fig. 2f, we have a resonant peak at lower frequency, i.e. closer to Gamma, corresponding to better matching between the probe and the sample at this frequency. Therefore, we decided to show this peak as an example of a s-mode.

R2: *The frequencies of the band gap in Fig. 4b and Fig. 4d are different. What is the main reason for this frequency shift?*

A: We are thankful to the referee for this interesting question. It is true that there is a small frequency shift between numerical simulation and experiments. On the one hand, we note that the experimentally measured bandgap spreads over a smaller frequency range than the simulated one. This is explained by the finite size of the transmission peaks which is due to the inherent losses and finite size of the sample. On the other hand, the dispersion relation displayed on Fig. 4d is calculated for a perfectly aligned configuration of the two lattices. Although we did our best to align the two

experimental samples, the measured spectrum presents some minor deviations compare to the numerical results. This also leads to the fact that the alignment of the two lattices is important for the dispersion of the edge mode in that sense that it is related to the breaking of the C_6 symmetry. These important details are added to the revised manuscript.

R2: *And how do the authors decide the nontrivial band gap region in Fig. 4b?*

A: We thank the reviewer for this question. The non-trivial bandgap region, represented in blue in Fig. 4b, corresponds to the frequency range where the experimentally measured trivial and non-trivial bandgap overlap. We stress this important point in the revised manuscript.

R2: *Minor issue:By the end of the introduction paragraph, the authors mentioned that “It certifies that metamaterial crystals can be engineered whose potential greatly surpass that of conventional ones”. This statement is vague or at least needs further interpretation.*

A: We thank the reviewer for this comment. As both reviewers suggested, we have rephrased the sentence and turned it into a more meaningful statement.

R2: *By the end of the first paragraph on page 4: “the first consequence of the crystalline nature of the metamaterial appears: the dispersion relation which links the frequency to a wavenumber is direction dependent.” While this sentence by itself is not wrong. However, this sentence is not consistent with the following sentence “mostly gives rise to isotropic properties”*

A: We thank the reviewer for raising this unclear point. What we meant is that generally speaking, the crystalline nature of the metamaterial implies that the frequency/wavenumber relation *can be* direction dependent. For simple arrangements, typical in metamaterials, like the one of the triangular lattice, the dispersion relation is mostly isotropic, and the structure does not induce any notable spatial dispersion. But when the structure is more complex, like the one of the honeycomb, the crystalline effects are very important to consider. In particular, here they explain the nontrivial topology. We make this point clearer on the revised version of the manuscript.

R2: *In the second paragraph on page 4, the authors change from triangular lattices to honeycomb lattices and then obtain Dirac cones. When I was reading this paragraph, I feel a little misleading as the triangular lattice also supports Dirac cones, but at a higher frequency. I understand that the authors are working in deep subwavelength region. However, I think it is better to point out this explicitly.*

A: We are thankful to the reviewer for this suggestion, and we have clarified this point in the main text. It is true that the triangular lattice also supports Dirac cones, but as the referee has also noted, those degeneracies lay within the light cone preventing the subwavelength nature of our study.

R2: *The first sentence on page 5. “Going from the triangular lattice to the honeycomb one can be*

pictured as a folding of the band structure, the area of the first Brillouin zone being divided by 3.” I guess here is a typo, from triangular lattice to honeycomb lattice, the first BZ is divided by 2 not 3.

A: We thank the reviewer for this comment. However, we believe that we are correct. Indeed the area A_T of the first Brillouin zone in the case of a triangular lattice with a step a is:

$$A_T = \left(\frac{2\pi}{3a}\right)^2 \sqrt{\frac{10}{3}}$$

The corresponding honeycomb lattice has a step $\sqrt{3}a$, therefore the corresponding area A_H is :

$$A_H = \frac{1}{3} \left(\frac{2\pi}{3a}\right)^2 \sqrt{\frac{10}{3}} = \frac{A_T}{3}$$

Hence the area of the first Brillouin zone is divided by three between the triangular and corresponding honeycomb lattice. This is also directly apparent in the honeycomb lattice Brillouin zone pictured in green in Fig. 1c, which is overlapped with the triangular lattice Brillouin zone in blue.

R2: *Two sentences below. “Thus, we consider the supercell containing a full hexagon of resonators instead of the primitive one containing only the 2 resonators (Fig. 1d)”. Here the citation of figures are not proper. I think it should be “Thus, we consider the supercell containing a full hexagon of resonators (Fig. 1d) instead of the primitive one containing only the 2 resonators (Fig. 1c)”*

A: We have fixed the issue in the revised manuscript.

R2: *In the middle of the first paragraph on page 12. “In stark contrast, leaky wave out of plane radiation is not allowed along the interface due to the subwavelength nature of the edge mode.” I get a little confused here. In the demonstration, the edge mode and nontrivial band gap is inside the light cone, hence I think it can couple with the light cone. Do I miss something here?*

A: We apologize for this inaccuracy in our language that we have fixed in the revised manuscript. LW radiation is obviously possible as soon as the edge mode enters the light cone. What we meant is that even in the light cone, the mode has a strong spatial Fourier component in the second Brillouin zone, since built on folded bands, and therefore the leaking rate for the edge mode is very small compared with the leaking rate at the end of the sample, which is the one associated with a matched dipole. Therefore, for short samples, radiation occurs mostly at the ends of the waveguide, allowing for efficient coupling to the far field. We have modified the paper accordingly, since we do not want the readers to be confused as well.

Reviewers' comments:

Reviewer #1 (Remarks to the Author):

All my comments have been addressed. The manuscript is ready for publication.

Reviewer #2 (Remarks to the Author):

The authors have successfully addressed most of my concerns in the last report. However, I still have a few further concerns.

1. Following the second question in the last report. I understand the tradeoff between localization and robustness. And I also agree that the robustness of the surface states can be preserved to some extent. However, I still think that the one-way property of the surface states at these two frequencies, HF and LF, is questionable.

Due to the breaking of the C_{6v} symmetry at the surface, there is a little gap between the surface bands in Fig. 4d. Am I correct that HF and LF represent the two band edge states of the surface bands? (Just to make sure I understand the caption correctly) If so, then a simple implication is that the group velocity of these two states are exactly zero, and then what are the one-way directions of these two states? (Note here due to spin hybridization, we cannot say different propagating directions for different spins)

Also in the reply, the authors mentioned that "of all the kinds of defects that we have tested, only the interface with air showed noticeable backscattering. We had to interface the crystal with a continuous medium to start destroying the one-way property." Is there a simple explanation for this phenomenon?

2. For the sixth question in the last report about the identification of mode profile. I agree with the authors that the modes are only well-defined near the Gamma point. The mode are not pure for any k points away from the Gamma points. Hence the mode profiles can only be approximately obtained. For s state, Gamma points is at zero frequency hence experimentally difficult to measure. But the following reason given by the authors is not that convincing: "At the same time, it is clear that the field is mostly of monopolar nature" It seems to me that the authors measured the field patterns for different frequencies and then pick one frequency of which the field pattern is mostly of monopolar nature. This doesn't make much sense. If coupling is the only issue, then the authors should choose the highest peak slightly below the "s" state in Fig. 3d. However, the field pattern for this frequency is not monopolar like? In my understanding, the main physics comes from the p and d orbitals, then the "s" orbital is not that important to show.

Reviewer #1: *All my comments have been addressed. The manuscript is ready for publication.*

Authors: We would like to thank the referee for his/her time and for his/her positive recommendation.

Reviewer #2: *The authors have successfully addressed most of my concerns in the last report. However, I still have a few further concerns. Following the second question in the last report. I understand the tradeoff between localization and robustness. And I also agree that the robustness of the surface states can be preserved to some extent.*

Authors: We are happy that the referee has grasped the main claims that we make in this section of our manuscript, and now agrees that the robustness of the edge states can be preserved to some extent, as we demonstrate with both simulations and measurements. Below, we respond to his/her other comments.

R2: *However, I still think that the one-way property of the surface states at these two frequencies, HF and LF, is questionable. Due to the breaking of the C_{6v} symmetry at the surface, there is a little gap between the surface bands in Fig. 4d. Am I correct that HF and LF represent the two band edge states of the surface bands? (Just to make sure I understand the caption correctly) If so, then a simple implication is that the group velocity of these two states are exactly zero, and then what are the one-way directions of these two states? (Note here due to spin hybridization, we cannot say different propagating directions for different spins)*

Also in the reply, the authors mentioned that “of all the kinds of defects that we have tested, only the interface with air showed noticeable backscattering. We had to interface the crystal with a continuous medium to start destroying the one-way property.” Is there a simple explanation for this phenomenon?

Authors: As the referee interprets and points out correctly, the one-way property of the states is not valid exactly at Γ , since the group velocity is zero due to the mini gap, and no direction can be defined. However, this does not put into question the one-way property. Indeed, slightly departing from Γ (i.e. at any other point) the slope is nonzero, so we can unambiguously define a direction of propagation and talk about the directionality of the mode. Our unique claim here is that in close vicinity of Γ , we have directional states that are extremely robust to backscattering. This claim is already fully demonstrated by the data in our paper, but we have stressed it better in the revised version.

As for the second part of the comment, we believe it comes from the fact that in the case of a continuous medium, and very different from the case of a C_6 -symmetric crystal, it is impossible to define two degenerate pseudo-spin states that are compatible with C_6 symmetry. Therefore, spin degeneracy is not valid in a continuous medium, and the analogy with a spin-Hall system, where both topological media support spin-degeneracy, cannot be made. We added a short sentence to stress this fact in the revised version.

R2: *For the sixth question in the last report about the identification of mode profile. I agree with the*

authors that the modes are only well-defined near the Gamma point. The mode are not pure for any k points away from the Gamma points. Hence the mode profiles can only be approximately obtained. For s state, Gamma points is at zero frequency hence experimentally difficult to measure. But the following reason given by the authors is not that convincing: "At the same time, it is clear that the field is mostly of monopolar nature" It seems to me that the authors measured the field patterns for different frequencies and then pick one frequency of which the field pattern is mostly of monopolar nature. This doesn't make much sense. If coupling is the only issue, then the authors should choose the highest peak slightly below the "s" state in Fig. 3d. However, the field pattern for this frequency is not monopolar like? In my understanding, the main physics comes from the p and d orbitals, then the "s" orbital is not that important to show.

Authors: We are thankful to the reviewer for this observation. We agree that most of the interesting physics in this study lies by the d and p "orbitals", so the comment is question about whether to show or not the field profile at lower frequencies. We would like to stress that there is nothing wrong with this field profile: we already clearly state that it is not taken at zero frequency, and that it features a s -dominated crystal mode, with a Bloch phase on top. Because we believe that showing a field map at low frequency is relevant to exemplify in the most complete way the "locally resonant metamaterial/Topological crystal" approach we present here, we would like to keep the manuscript as it is.

That being said, we understand from the referees comment that our previous response was confusing. The referee asked how we chose the frequency for the s mode, and we previously replied that we chose a frequency at which coupling was efficient. This apparently confused the referee into thinking that only at this frequency the field was of monopolar nature. This is not the case. Indeed, looking at the field maps at other low frequencies, we see the same type of s -dominated mode, so the choice of the frequency is unimportant. We apologize for the confusion.